# Livelihood Capitals and Opportunity Cost for Grazing Areas’ Restoration: A Sustainable Intensification Strategy in the Ecuadorian Amazon

**DOI:** 10.3390/ani13040714

**Published:** 2023-02-17

**Authors:** Bolier Torres, Ítalo Espinoza, Alexandra Torres, Robinson Herrera-Feijoo, Marcelo Luna, Antón García

**Affiliations:** 1Faculty of Life Sciences, Amazon State University (UEA), Pastaza 160101, Ecuador; 2Department of Animal Production, Faculty of Veterinary Sciences, University of Cordoba, 14071 Cordoba, Spain; 3Postgraduate Unit, State Technical University of Quevedo (UTEQ), Quevedo Av. Quito km, 1 1/2 Vía a Santo Domingo de los Tsáchilas, Quevedo 120550, Ecuador; 4Faculty of Biological Sciences, State Technical University of Quevedo (UTEQ), Quevedo Av. Quito km, 1 1/2 Vía a Santo Domingo de los Tsáchilas, Quevedo 120550, Ecuador; 5Faculty of Legal, Social and Education Sciences, Technical University of Babahoyo (UTB), Km 3 1/2 Vía a Valencia, Quevedo 120550, Ecuador; 6Faculty of Agriculture and Forestry, State Technical University of Quevedo (UTEQ), Quevedo Av. Quito km, 1 1/2 Vía a Santo Domingo de los Tsáchilas, Quevedo 120550, Ecuador; 7Faculty of Earth Sciences, Amazon State University (UEA), Pastaza 160101, Ecuador

**Keywords:** Sumaco Biosphere Reserve, cattle income, land sparing, land sharing

## Abstract

**Simple Summary:**

Land sparing and land sharing should be considered complementary strategies to favor pasture areas’ restoration in the Ecuadorian Amazon Region (EAR). Their implementation will depend on factors such as the livelihoods, natural resources’ valuation and income from livestock activity (dual-purpose cattle farms). A sample of 167 farms of the EAR distributed in the three altitudinal gradients (high, medium, and low) was used. The different livelihood capitals and the opportunity cost of the grazing area was calculated. Therefore, to promote pasture restoration areas, policy strategy should aim to maximize net benefits by unit area (ha). Starting from the results obtained, different strategies of restoration are proposed by altitudinal gradient and productive specialization: (a) Land-sparing restoration actions; in middle hill and low zones, land could be destined to produce ecosystem services, i.e., a strategy that promotes ecosystem restoration through environmental payments and considers at least the opportunity and transaction costs. (b) Land-sharing pasture restoration, which would focus on high mountain farms with a strategy of sustainable intensification and food security.

**Abstract:**

Land use change in pastures is considered one of the leading drivers of tropical deforestation in the Ecuadorian Amazon Region (EAR). To halt and reverse this process, it is necessary to understand, among other factors, the local livelihoods, income from grazing area and the appropriate options to foster sustainable production, incorporating the land-sparing and land-sharing approach. This work was conducted using 167 household surveys along an altitudinal gradient within the buffer and transition zone of the Sumaco Biosphere Reserve (SBR) in the EAR. The results of a comparative analysis of the main capital variables (human, social, natural, financial, and physical), and the opportunity cost of grazing area assessment provides the following key findings: (a) the concepts of land sparing and land sharing should be considered as complementary local strategies, including household livelihoods and the opportunity cost of the grazing area; (b) we should encourage markets with differentiated restoration rights, based on households engaged in low grazing areas’ opportunity costs, and making less impact on capitals’ livelihood a key element of economic and conservation initiatives; and (c) sectoral policy implications, including moderate intensification and technological improvements to strengthen the pastureland-sparing and -sharing approach, are discussed.

## 1. Introduction

Cattle ranching has a fundamental role to play in rural households’ livelihoods, and in the economy of households from developing countries [1,2,3]. This activity is considered a source of future capital savings for rural households, allowing them to generate income by trading animals in times of economic uncertainty [4,5,6]. In this context, for example, beef exports exceeded 153.1 billion lbs worldwide in 2018. Of this value, 25.7% was produced in Latin America [7,8]. In tropical areas, pastures are the main source of feed for ruminants, which has encouraged farmers to implement large extensive livestock grazing [9,10] to meet the feed demand of these livestock [11]. However, these management systems are associated with major factors of tropical deforestation, soil degradation and local biodiversity loss. [12,13]. In addition, it is also estimated that livestock farming contributes approximately 15% of global greenhouse gas emissions [14,15,16].

The world faces the challenge to encourage best management practices (BMPs) for livestock-oriented activities [4] to mitigate climate change [17], given that cattle-raising activities are the principal cause of changes in global land use [18]. In this sense, the shifting use of land intended for pastoral purposes has been very intense in recent decades [19,20], with about 178 million hectares (ha) of forest being lost from 1990 to 2020, at a rate of 7–8 million ha/year in the period 1990–2000, 5.2 million ha/year in 2000–2010 and 4.7 million between 2010–2020 [21]. Currently, approximately 37% of the global area is known to be under livestock grazing activities [22].

This contribution was carried out in the Ecuadorian Amazon Region (EAR), where the agricultural frontier is expanding at an unprecedented rate [23,24], with cattle pastures being one of the main drivers of land use change and deforestation [24,25,26]. It is estimated that in this region, in 2014, there were approximately 1.2 million hectares of pasture for extensive livestock grazing [9], contributing to meeting the national and international demand for animal protein and milk [27]. On the other hand, despite the low income from agricultural activities in general in the rural EAR, the livestock-based livelihood strategy is the most economically profitable source of income for households compared to other strategies, based on crops and forestry activities at household level [1,28]. Particularly, in the SBR, it is estimated that 56.1% of the productive activities carried out by local settlers are based on subsistence methods focused on cattle ranching [4].

The Ecuadorian government launched the Amazonian Productive Transformation Agenda—(ATPA, for its Spanish acronym). ATPA intended to convert 300,000 hectares of pasture into silvopastoral systems (SPS), integrated forestry and mixed agroforestry systems [29]. This initiative is aligned with other global restoration efforts such as the Bonn Challenge [30] and the UN Decade of Restoration [31], which aim to reforest 350 million hectares by 2030 [32,33]. In this context, a growing interest is evident in the search for various forms of financing at the global level to implement these practices [34,35,36]. However, it is necessary to explore appropriate approaches to successfully implement these actions at the local level. Within this framework, this research aimed to propose a strategic program to promote sustainable intensification in the Ecuadorian Amazon with the following steps: (a) identify and access livelihood capitals along the altitudinal gradient; (b) estimate the opportunity cost of the grazing area (OppCost_grass_ha_) through the cost–benefit ratio/grazing area (ha), and (c) design strategic measures for sectoral policy mix between land-sparing and land-sharing approaches, according to ethnicity and gradient range.

## 2. Theoretical Framework

Three main concepts involving households’ decisions to suggest livestock sustainable intensification strategies oriented to land-sparing and land-sharing approaches along the altitudinal gradient studied were used. 

Firstly, we use the sustainable livelihood framework (SLF) as an appropriate theoretical approach that integrates concepts of development and conservation [37,38,39] considering household as the main factor of farm-activity decision making, and also provides policymakers with specific information to design sustainable development and restoration policies. This approach has been used by previous studies focused on livelihood strategies (LS) in rural areas [28,40,41,42,43] at household level. this approach allows us to analyze both assets (human, social, natural, physical and financial) and external factors that households use in their on- and off-farm activities for survival and for improving standards of living [44,45,46,47].

Secondly, opportunity cost was used for the indirect valuation of ecosystem services and their subsequent use for developing public policies and payments for environmental services (PES) programs [48,49]. The direct opportunity cost of the productive activity is the REDD+ valuation method for reducing greenhouse gas emissions (GHG) from deforestation and degradation [50,51]. Thus, the direct use value (milk and meat) competes with the conservation of ecosystems and the mitigation of climate change; therefore, land use is linked to the objectives of sustainable development and poverty eradication, whilst aiming for ecosystem stability [52,53].

Diverse methodologies exist for ecosystem service valuation [48,49]. Direct methods use market prices of the commodities generated (milk and meat), but in the absence of direct environmental economic values, opportunity cost was considered a robust and reliable indicator for the valuation of ecosystem services. The opportunity cost compares the sacrificed benefits from land use with the environmental services that could be provided [54,55]. According to Leguía & Moscoso [56], the REDD+ opportunity cost methodology can be extended to other cases and ecosystem services. Thus, we used the adapted opportunity cost (the net benefit of the grazing area by hectare) as a complementary approach, given that cattle production is the main driver of tropical deforestation, causing greenhouse gas emissions on both a regional and global scale [57]. Therefore, implementing mechanisms for landscape restoration is a top priority within climate change mitigation efforts, wherein countries should encourage actions that promote the reduction of greenhouse gases (GHG). One option is to promote restoration processes of extensive pasture systems; however, in this case, households would face financial losses due to decreased income from their grazing activities, which could be compensated through mechanisms such as restoration incentives that could be estimated through the opportunity cost of the grazing area in different zones. This approach recognizes that in the absence of incentive and compensation mechanisms, these losses will have to be absorbed by the households themselves [58]. Furthermore, although a suitable political climate to promote compensation mechanisms for restoration is required, appropriate mechanisms for small producers need to be estimated as tools to facilitate their implementation [59].

Additionally, scientific evidence suggests that sustainable intensification coupled with technological improvements could meet future food demands while avoiding the emissions typically associated with deforestation [60]. Along these lines, several authors recognize two sustainable intensification strategies: first, land sparing, which involves land preservation or the idea of intensifying productivity in one area through higher yields, while conserving another [61,62,63]; the second is land sharing, or wildlife-friendly farming [64], wherein livestock itself can foster ecosystem services through the landscape, triggering ecological [65] and productivity gains in a landscape [66,67].

These three approaches are appropriate for the EAR, where unsuitable cattle-raising practices [68,69] are threatening the conservation and sustainable use of natural resources. Thus, cattle landscape restoration could be promoted, using these three approaches as the conceptual framework for this study (Figure 1).

## 3. Materials and Methods

### 3.1. Study Area

The study area includes the Sumaco Biosphere Reserve (SBR) located in the provinces of Napo (62%), Orellana (35%) and Sucumbíos (3%). The study communities are located from 400 m asl in the lower zone to 2000 m asl in the upper zone of the SBR (Figure 2). In addition, the study area is part of the Uplands Western Amazonia biodiversity hotspot [70].

### 3.2. Pastoral Context

Studies conducted by Torres et al. [71] show that farm size means along the gradient (Figure 3) denote in the medium zone a typical farm with 62.4 hectares, distributed as follows: 55% for growing pastures, 40% covered by forest, and the remainder dedicated to crops. In the lower zone, the average size per farm was 47.3 ha, with 62% pastures, 34% forest and 5% crops. In the highland, the land (35.2 ha/farm) was mostly to produce pastures (81%), whereas 17% was covered by forest and less than 2% was used for crop purposes. The productive objective of the farms in the three gradients was focused on raising cattle (most strongly in the high and low zones), while in the medium zone, much of the land was covered by forest. However, the availability of farmland in the three zones is very low.

### 3.3. Sampling and Data Collection

The study was performed in the buffer and transition zone of the SBR. From 464 households distributed in the three zones, those farms with more than ten heads of cattle and more than three years of consecutive dual-purpose activity (both milk and meat) were identified. Dual-purpose cattle is a mixed system widely distributed in developing countries, characterized by small size, low productivity, little or no technological level, low-level of use of external inputs and diversified activities (milk, meat, work, crops, etc.) and a high level of marginalization of the farmers [2,5,49,72].

With this database, a stratified non-experimental design was applied, controlled by the effects of climatic conditions and altitudinal ranges, to determine differences in the edaphological and climatic characteristics of the landscape. In this way, 167 households were interviewed by stratified randomized sampling, with proportional assignation along elevational gradient of the study area (Figure 1); there were 57 farms in the low zone (Carlos J. Arosemena Tola canton), 57 in the medium zone in Archidona canton (Cotundo), and 53 in the high gradient in Quijos canton (San Francisco de Borja). All farms were located in the SBR.

### 3.4. Computing Cattle Ranching Income and Cost

For the calculation of cattle costs and income, we considered the fixed costs of land rent, maintenance of facilities or amortizations and financial expenses, and also the variable costs of purchase of fattening cattle, various inputs, and maintenance of pastures. The sum of the fixed and variable costs determined the total cost per household [73,74]. The net profit per household was obtained from net income minus total costs. The benefit–cost ratio was obtained from the sum of net income divided by total costs, using the following formula: (1)$$B/C_{\mathrm{cattle}\_\mathrm{ha}}=\overset{N}{\underset{i=1}{\sum }}\left[\frac{\;\mathrm{income}\_{\mathrm{ha}}_{i}\;}{\cos t\_{\mathrm{ha}}_{i}}\right]$$

From the 167 cases (lowland, middle hill, and high mountain zones) 37 cases reported obtaining credit (86% by the state bank) of which 11 are from the lowland zone with an average of USD 7636 in credit received, 9 are from the middle hill zone with an average of USD 8816, and 17 are from the high mountain zone, with an average of USD 13,480. This shows that those in the high mountain zone had an important flow of financial capital to their activity, which can be seen in the benefit/cost ratio index. Among these cases, 31 reported a monthly payment value: 6 from the lowland zone, with an average of USD 427.9 from the middle hill zone, with an average monthly payment of USD 188.26, and 16 from the high mountain zone, with an average of USD 944.

### 3.5. Computing Grazing Area Opportunity Cost to Promote Restoration

To measure the opportunity cost of the grazing area in hectares (OpptCost_grass_ha_), we first used the net benefit, which is the result of income minus the cost of the cattle ranching activity by hectare (income__ha_ − cost__ha_). The complete formula applied in this paper is described as follows:(2)$${\mathrm{OpptCost}}_{\mathrm{grass}\_\mathrm{ha}}=\overset{N}{\underset{i=1}{\sum }}\left[\frac{\left(\mathrm{income}\_{\mathrm{ha}}_{i}\right)-\left(\cos t\_{\mathrm{ha}}_{i}\right)\;}{\mathrm{grass}\_{\mathrm{ha}}_{i}}\right]$$
where N is the number of cattle ranching households, income__hai_ is the total household income from activity i (milk and meat production), cost__hai_ is the total cost of cattle ranching activities and grass__hai_ is the total grazing area in hectares. We found the annual opportunity cost for each hectare of pasture that a farmer has over along the altitudinal gradient studied, considering the net benefit that a producer will renounce for each hectare of pasture released for restoration.

## 4. Results

### 4.1. Cattle Strategies, Ethnicities and Location

Cattle ranching systems were established at different times along the altitudinal gradient throughout the current Sumaco Biosphere Reserve, which corresponds to the Napo province (Table 1). Thus, ranchers initially colonized the high mountain zone (1601–2000 masl) 70 years ago, followed by the lowland zone (400–700 masl) approximately 47 years ago, and then they settled in the middle hill zone (701–1600 masl) about 38 years ago (Figure 2). Only in the middle hill zone were indigenous cattle ranchers registered. In this zone, 56.1% of the cattle ranchers are of the Kichwa nationality, and have adopted a subsistence strategy based on cattle ranching. Conversely, in both the lowland and high mountain zones, only mestizo cattle ranchers were recorded (Table 1).

Cattle raising strategies used in both the lower and middle zones are focused on meat and milk sales, while only in the high mountain zone are the activities associated with milk production. With respect to the feeding system, grasses were predominant in comparison to leguminous plants. In this context, the high mountain zone was highlighted to have 91.7% of grasses, followed by 86.7% in the middle hill zone and 60% in the lowland zone. However, the percentage of legumes was higher in the lowland zone, with 40%, while the middle hill had 13.3%, and only 8.3% was present in the high mountain zone. A wide description of the productive systems in the studied area can be found in Torres et al. [4,71].

### 4.2. Main Livelihood Capitals

#### 4.2.1. Human Capital

The size of cattle-raising households throughout the studied gradient was from 5.04 to 6.70 household members, with an average of 5.78. The age of the head of household was over 54 years old. However, it was found that an average of 2.66 household members work directly in livestock activities as their main source of income. An average of 2.59 household members work in off-farm activities. It was also observed that cattle ranchers in the middle hills have the highest percentage of illiteracy (15.8%), compared to the lowlands (8.8) and high mountains zone (3.8). The highest levels of secondary and university education were observed among households in the high mountains (Table 2).

#### 4.2.2. Social Capital

With regard to social capital, an average of 51.5% of the households (in all three zones) belong to a producers’ association, with the highest percentage of associates in the middle hills zone (61.4%). On the other hand, an average of 46.7% of the associates participate actively, and 85% feel satisfied with their livestock activities. Furthermore, it is important to recognize that in the middle zone, almost 80% of the households have replacement generations (household members interested in continuing with these activities), while in the low and high zones, only 56% and 64% have been found, respectively.

#### 4.2.3. Natural Capital

The results show an average farm area of 48.59 ha, with range of approximately 35 to 62 ha for all the cattle ranching households sampled. However, the largest pasture area was found in the middle hill zone, with a mean of 27.20 ha, followed by the lowland zone, with 26.81 ha, and the high mountain zone, with 22.52 ha, without significant differences along the gradient analyzed. In addition, it was found that in the high zone, around 80% of the pasture areas are compatible with grazing. On the other hand, the average percentage of forested land within the cattle farms was higher in the middle hill zone with 33 ha, compared to 20 ha and 12 ha for the lowlands and high mountain zone, respectively. Crop areas of these producers are reduced, with an overall average of 1.41 ha of land under agricultural crops.

#### 4.2.4. Physical and Financial Capital

In this regard, it was found that the high mountain zone has a larger percentage (47.20%) of farmers specializing in dairy farming. It was also found that 37.70% of the farmers in this zone count on appropriate infrastructure and animal facilities.

In terms of financial capital, the results indicate an average of 15.6% of farmers in the three zones analyzed have access to credit to develop productive activities. However, the animal stock and average annual milk production differ significantly, with the high mountain zone having the highest average number of animals (30.4 heads) and the highest average annual milk production yield (32,654.21 l/farm and year L/year). This yield is related to the investment because the greatest annual investment was identified in the high zone, at $4307.38.

### 4.3. Opportunity Cost of Grazing Area

The results show an average gross annual income of USD 8152.03 from meat and milk sales along the altitudinal gradient studied, with the high mountain zone having the highest productivity with values above USD 19,042 (Table 3). Within this context, it was evident that there is greater profitability in the high mountain zone, with a net profit of USD 14,735.36, compared to USD 1859.32 in the middle hill zone and USD 1052.77 in the lowland altitudinal gradient. Moreover, the average benefit–cost ratio was USD 3.23 in all zones; however, this ratio was higher in the high mountain zone, at USD 5.21. Finally, the opportunity cost of the grazing area was significantly higher in the high mountain zone, at USD 672.36, followed by USD 58.85 for farmers located in the middle hill zone and USD 37.08 for farmers located in the lowland altitudinal gradient.

## 5. Discussion

### 5.1. Livelihood Capital and Opportunity Cost for Grazing Areas Restoration

Pasture systems are considered the main drivers of tropical deforestation [24,25,75], especially because of their extensive use due to the low net income per hectare obtained by producers in some areas [4,76]. Therefore, the promotion of public policies to encourage the restoration of ecosystems in the Amazon through economic valuation of ecosystem services should be of particular interest [12,77,78]. In addition, payments for environmental services could be linked to the development of sustainable intensification practices [79,80,81] aimed at promoting restoration in agricultural and livestock systems. In this research, according to Reyes [48], Börner and Wunder [82], and Leguia and Moscoso [56], opportunity cost was quantified through the valuation of land use and the provision of market products (meat and milk). The methodology was adapted according to the livelihood capitals, production systems and the grazing area from an altitudinal gradient in a zone of high diversity and endemism in the EAR [83,84,85,86].

The findings suggest that the lowland and middle zones are appropriate for promoting restoration, considering the high natural capital (including grazing areas) in these zones and the particularly low opportunity cost of each hectare of pasture, with averages of USD 37.08 and USD 58.85 per year, respectively; meanwhile, in the high mountain zone, the opportunity cost of the grazing area had an average of USD 672.36 per year, where any compensation option could be less attractive, challenging and expensive (Figure 4). In this respect, we support the use of opportunity cost to establish environmental payment policies and development of BMPs for sustainable intensification [48,56,74].

Considering the theory of natural resource valuation [54,87], based on these results, the impact of land use change is valued and contributes to the planning of its optimal use and the development of programs for payments for environmental services. Figure 4 shows the variation of the opportunity cost according to altitudinal gradient and grazing area. In this regard, Reyes [48] indicated that agricultural productivity responds in a manner inversely proportional to the slope of the altitudinal gradient; thus, the higher the gradient, the lower the opportunity cost of the land. Our results presented opposite patterns; the zones in the highest gradient specialized in dairy production and showed a high opportunity cost. On the contrary, as the altitudinal gradient decreased, the systems evolved to dual-purpose (meat-milk) in an extensive way, presenting a lower stocking rate and lower opportunity cost.

If the opportunity cost and capital variables vary along the altitudinal gradient, these results could be used to optimize land use and promote sustainable development goals (SDGs) and the REDD+ methodology [4,48,56,71]. Thus, considering the opportunity cost curve and the natural capital variable related to pasture area (Figure 4), in high mountain zones, the systems could be focused on food production contributing to (SDG 1), using the land-sharing approach. On the contrary, in the middle hill and lowland zones, with low opportunity cost, the land could be destined for the production of ecosystem services (SDG 13, SDG 15) based on the land-sparing approach. The combination of these two strategies favors sustainability [88], and is compatible with the achievement of the goals of the 2030 agenda (SDG 13, SDG 15) [54,86,89]. However, a limitation of the use of opportunity cost in the valuation of ecosystem services is the variability of productivity and prices over time, so it is necessary to promote longitudinal studies or, alternatively, probability models that simulate the behavior of these parameters.

#### 5.1.1. Opportunity Cost to Release Pasture Areas for Restoration (Land Sparing)

The results using the livelihood and opportunity cost approaches of the grazing area suggest the strategy of the land-sparing approach, which is oriented to release pasture areas for the protection of local biodiversity through active or passive restoration processes. This is recommended mainly in the lowland and middle hill zones (Figure 4), due to the low opportunity cost that makes it easier for producers to accept the initiation of restoration processes if human, natural and financial capitals are also valued [1]. Therefore, the plantation of forests could contribute to better management of the productive landscape, especially in highly degraded areas, as well as livelihoods benefits [90].

Beneath this approach, the implementation of BMPs oriented to pasture area restoration could be financially promoted through an incentive program [4]. Larger incentives and a “Chakra” system (traditional agroforestry system characterized by its high levels of timber and fruit trees content), moderate incentives for activities promoting active and passive restoration, and smaller incentives for agroforestry practice are suggested for reforest. Likewise, operational investments need to be calculated, which would be added to the opportunity cost; thus, its minimum value would be the cost associated with the adoption of the BMPs [91].

In particular, this strategy facilitates the creation of incentives to promote and enable the sustainable intensification of livestock production in small areas, converting such areas into highly profitable areas, allowing: (a) restoration of degraded areas in livestock landscapes, thereby conserving the biodiversity surrounding the protected areas that are considered a hotspot of biodiversity and endemism [70,92], contributing to the stability and resilience of the ecosystem and consequently to a decrease in GHG emissions and carbon sequestration [17] from cattle production. (b) This approach could also contribute to reduciong deforestation and habitat fragmentation in an area of high diversity [93,94], which is essential to avoid species loss; (c) the land-sparing approach and complementary public policies such as incentives, technical assistance, and access to flexible credit would allow greater efficiency [94,95] in this case in livestock production, since both the release areas for restoration and the areas destined for production could be used intensively, maximizing their yields, to benefit the households and the landscape. These processes could help to meet the growing demand for food from deforestation-free areas in an increasingly populated world [96,97,98] with sustainable development objectives.

#### 5.1.2. Opportunity Cost for Cattle Sustainable Intensification (Land Sharing)

Land sharing could be a suitable option for the Amazon region. Local producers have planted trees or allowed trees to be regenerated naturally in pastures and in combination with other agricultural plants and fruit trees, thereby creating small agroforestry clusters and tree nucleation around grazing areas which act as green islands in open degraded pastures with more than 100 species [99], reducing the areas of monoculture pastures not only by intensification, as suggested by Green et al. [61] and Phalan et al. [100], but also avoiding the soil degradation.

Our results indicated that land sharing can be implemented in the three study zones (Figure 4), albeit with different strategies. According to Torres et al. [4], this could be the case for the three zones with grassland rehabilitation; this would be achieved with higher incentives in BMPs oriented to planting new trees in pastures, the establishment of tree cores around grazing areas, and lower incentives for the establishment of live fences surrounding pasture areas. Additionally, in high mountain areas, the objective should be to improve livestock with BMPs, through methods such as farm planning, establishment of accounting registers, establishment of compensation areas, establishment of enclosures with sheds, and improvement of the animal diet with mineral salts and dietary supplements. Due to the pasture areas in this zone, producers are more efficient in terms of income and livelihoods, so they would not be very motivated to release areas for restoration, given that any type of incentive would have to be equal or greater than the USD 672.36 that they receive per hectare annually.

### 5.2. Opportunities for Sectoral Policy Development

In this study, the opportunity cost for restoration appears as the loss of income of reducing grazing areas, as demonstrated by several studies that have identified more efficient and adaptable conservation strategies with this approach [101,102,103]. The land-sparing and land-sharing strategies facilitate reflection and to promotion of incentive policies as compensation for those who decide to restore or intensify their agricultural systems [104], thus encouraging producers to accept such policies, and also improving ecosystem functions towards the goals of not exceeding a 1.5 °C temperature increase [105,106] and contributing to the SDGs [71,107]. Simultaneously, the opportunity cost method can also be useful for designing research strategies for the adoption of BMPs [4,10], as well as for the adoption of new technologies [108,109] and the analysis of social networks in cattle smallholders that are involved in restoration processes [73] which can themselves be promoted through the enactment of a portfolio of incentives through sectoral policies or new political frameworks, thereby encouraging productive restoration [79,110,111].

Thus, to promote pasture restoration areas [77,78], policy strategy should aim to maximize net benefits by unit area (ha) and increase livelihoods [112,113,114]. This could be achieved by estimating the opportunity cost of the grazing area which, as evidenced in this study, even in the same region is not homogeneous, indicating that any type of intervention must be treated in a differentiated way to facilitate public policy. This is relevant, considering that under the Paris Agreement, countries have to submit their Nationally Determined Contributions (NDCs) to the United Nations Framework Convention on Climate Change (UNFCCC) [115]. In the case of Ecuador, the country submitted its first NDC in 2019 [116]; it included mitigation and adaptation plans determined in the National Climate Change Strategy (ENCC, for its Spanish acronym) 2012–2025 [117]. The ENCC includes land use change generated by the expansion of the agricultural frontier, with initiatives quantifying the potential for climate change mitigation through methods such as livestock restoration and intensification. All these restoration mechanisms would be more effective if only the opportunity costs of the pasture area were considered and treated in a differentiated manner.

In terms of research policies, these processes require monitoring not only in the adoption of new policies or technologies, but also in opportunity cost modeling at the level of species appropriate for restoration, predictions of possible environmental changes, and optimization of land use oriented to climate risks. Finally, agricultural lands are also at the frontier of deforestation [118,119]; however, these results support other studies suggesting that moderate intensification coupled with technological improvements could meet future food demands while avoiding the emissions typically associated with land clearing [52,60,120].

## 6. Conclusions

The study shows that opportunity cost, linked to net benefit from grazing area (ha), was a robust, direct, and simple and appropriate indicator. It was used to value the indirect use and hence the ecosystem services of ecosystem restoration. Thus, the opportunity cost could be used as a method to determine public policies for environmental payments and the development of best management practices for sustainable intensification. It will be possible to use these research results in a quantitative approach to environmental payments. Ecosystem services should incorporate, as a minimum, opportunity cost plus transaction and additional costs. These aspects could be considered in future research and represent a limitation of the study.

Two differentiated strategies are proposed: BMPs in land sparing are oriented fundamentally in middle hill and lowland zones, promoting ecosystem restoration through environmental payments. Furthermore, BMPs in land sharing would focus on high mountain farms with a strategy of sustainable intensification and food security.

However, both strategies should complement actions to concentrate conservation efforts, especially in areas of high biodiversity (hotspots), conserving ecological dynamics as much as possible through restoration strategies where they are most appropriate, and considering the livelihood outcomes and opportunity cost of grazing areas in pastoral regions.

## Figures and Tables

**Figure 1 animals-13-00714-f001:**
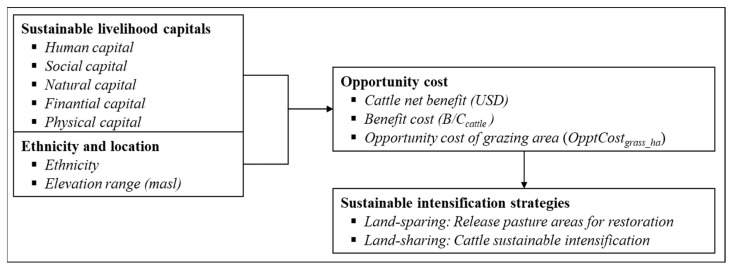
Conceptual framework.

**Figure 2 animals-13-00714-f002:**
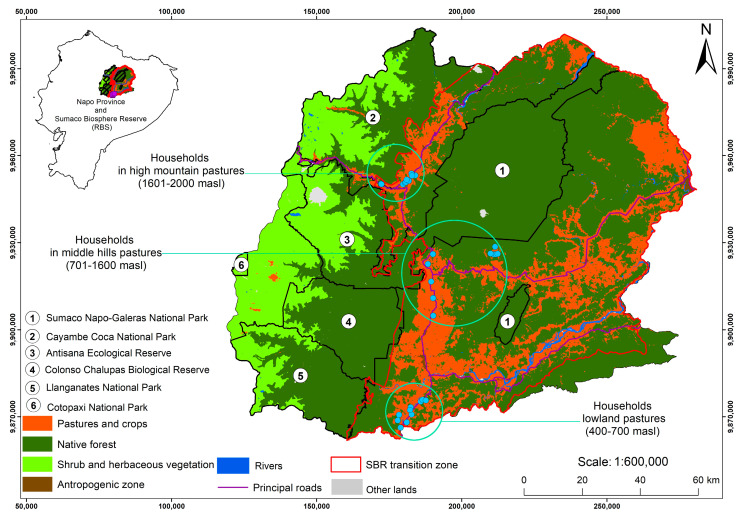
Distribution of farms in the three altitudinal zones, in the buffer and transition zone of the Sumaco Biosphere Reserve (SBR), Napo, Ecuador.

**Figure 3 animals-13-00714-f003:**
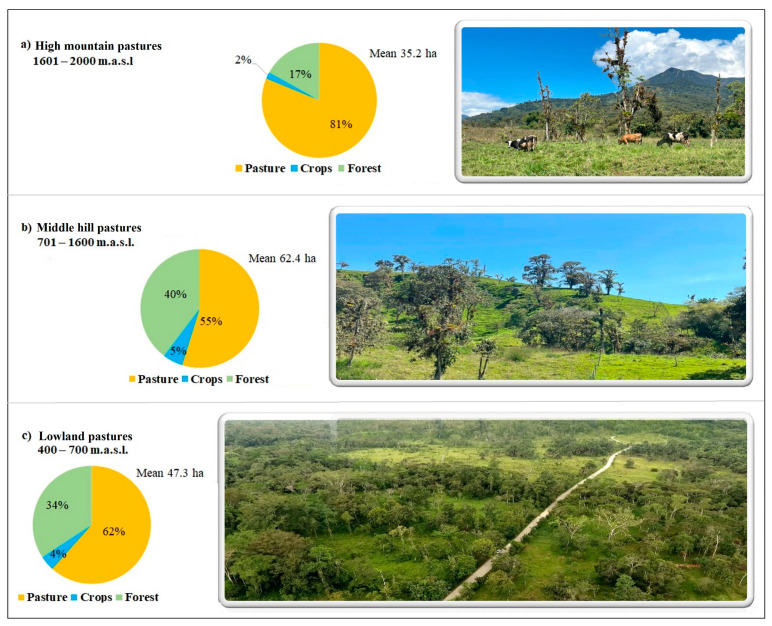
Mean of land use and farm size along altitudinal gradients pastoral systems: (**a**) high mountain zone; (**b**) middle hill zone; (**c**) lowland zone in Napo, Ecuador.

**Figure 4 animals-13-00714-f004:**
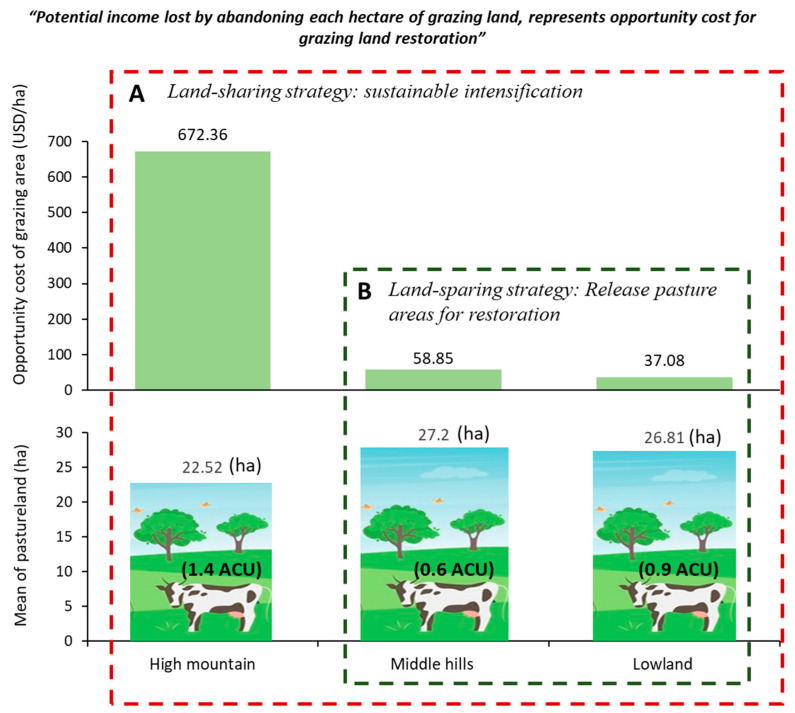
Opportunity cost of grazing area, represented by potential income lost by abandoning one hectare of grazing area. Additionally, suggested intensification strategies: (**A**) Potential for land-sharing and (**B**) for land-sparing approach along altitudinal gradients pastoral systems. Napo, Ecuador. (ACU is the Adult Cattle Unit).

**Table 1 animals-13-00714-t001:** Characteristics of cattle ranching farms along altitudinal gradients, Napo, SBR, Ecuadorian Amazon, 2015.

Variables	Altitudinal Gradients			Overall (n = 167)	*p*-Value ^1^
	Lowlands (n = 57)	Middle Hills (n = 57)	High Mountains (n = 53)		
Ethnicity and location					
Ethnicity (% Kichwa)	0.0 ^a^	56.1 ^b^	0.0 ^a^	19.2	***
Elevation range (m.a.s.l.)	400–700	701–1600	1601–2000	800	n/s
Average elevation (m.a.s.l)	543.11 ^a^	1114.16 ^b^	1778.02 ^c^	1129.93	***
Year of settlement (farm)	1975	1984	1952	1971	n/s
Slope in pastures (%)	26.81 ^a^	38.79 ^b^	24.91 ^a^	30.29	***
Cattle strategies					
Dual-purpose cattle system	Meat–dairy	Meat and dairy Dairy–meat	Dairy	Meat and dairy	n/s
Forrage types					
Grasses (%)	60	86.7	91.7		*
Legumes (%)	40	13.3	8.3		*

^1^*p*-Value: * *p* < 0.05; *** *p* < 0.001; n/s = not significantly differences between elevations gradients from 400 to 2000 m.a.s.l; ^a,b,c^ Letters in superscript denote significant differences among the three gradients.

**Table 2 animals-13-00714-t002:** Mean of the main capitals (human, social, natural, financial and physical) of cattle ranching systems along altitudinal gradients, Napo, SBR, Ecuadorian Amazon, 2015.

Variables	Altitudinal Gradients			Overall (n = 167)	*p*-Value ^1^
	Lowlands (n = 57)	Middle Hills (n = 57)	High Mountains (n = 53)		
Human capital					
Household size (n)	5.56 ^a,b^	6.70 ^a^	5.04 ^b^	5.78	**
Household members working on farm (n)	2.63	3.00	2.32	2.66	n/s
Household members working off farm (n)	2.12	2.68	3.00	2.59	n/s
Age of household head (y)	54.79	56.77	57.60	56.36	n/s
Household head without formal education	8.8	15.8	3.8	9.6	n/s
Household head with elementary school	61.4	47.4	28.3	46.1	n/s
Household head with secondary school	22.8	24.6	49.1	31.7	n/s
Household head with university level education	7.0	12.3	17.0	12.0	n/s
Social capital					
Member of association (Yes, %)	45.6	61.4	47.2	51.5	n/s
Actively participates in association (Yes, %)	45.6	54.4	39.6	46.7	n/s
Feel satisfaction on the farm (Yes, %)	87.7	82.5	84.9	85.0	n/s
Replacement generation (Yes, %)	56.1 ^a^	78.9 ^b^	56.6 ^a^	64.1	*
Production certification (Yes, %)	3.5	1.8	1.9	2.4	n/s
Natural capital					
Total land (ha)	47.30 ^a,b^	62.36 ^a^	35.16 ^b^	48.59	*
Pasture land (ha)	26.81	27.20	22.52	25.58	n/s
Pasture land compatible with grazing (%)	75.00 ^a^	54.12 ^b^	78.21 ^a^	68.89	***
Total forest land (ha)	20.01 ^a,b^	32.99 ^a^	12.16 ^b^	21.95	*
Total agricultural land (ha)	1.64 ^a^	2.17 ^a^	0.35 ^b^	1.41	***
Financial capital					
Access to credit for cattle system (Yes, %)	8.8 ^a^	14.0 ^a,b^	24.5 ^b^	15.6	*
Annual investment in cattle farm (USD)	1709.96 ^b^	1555.81 ^b^	4307.38 ^a^	2481.68	***
Physical capital					
Has cattle infrastructure (Yes, %)	8.80 ^a^	15.80 ^a^	47.20 ^b^	23.40	***
Infrastructure with healthy animals (Yes, %)	3.50 ^a^	7.00 ^a^	37.70 ^b^	15.60	***
Total stock of cattle (heads)	24.2 5 ^a^	18.84 ^a,b^	30.43 ^b^	24.4	**
Cows in production (heads)	12.36 ^a,b^	8.47 ^a^	15.08 ^b^	11.9	**
Productivity (l/farm and year)	1926.40 ^a^	2720.46 ^a^	32,654.21 ^b^	11,949.3	***

^1^*p*-Value: * *p* < 0.05; ** *p* < 0.01; *** *p* < 0.001; n/s = no significant differences between elevations gradients from 400 to 2000 m.a.s.l; ^a,b^ Letters in superscript denote significant differences among altitudinal gradients.

**Table 3 animals-13-00714-t003:** Mean of annual gross and net incomes; and opportunity cost of grazing area along altitudinal gradients, Napo, SBR, Ecuadorian Amazon, 2015.

Variables	Altitudinal Gradients			Overall (n = 167)	*p*-Value ^1^
	Lowlands (n = 57)	Medium Hills (n = 57)	High Mountains (n = 53)		
Gross income of meat and dairy (USD/farm)	2762.71 ^b^	3415.02 ^b^	19,042.63 ^a^	8152.03	***
Net benefit (USD/farm)	1052.77 ^b^	1859.32 ^b^	14,735.36 ^a^	5670.44	***
Benefit cost rate B/C_cattle_	2.29 ^b^	2.34 ^b^	5.21 ^a^	3.23	**
Opportunity cost of grazing area (OpptCost_grass_ha_) (USD/ha)	37.08 ^a^ (±145.1)	58.85 ^a^ (±189.2)	672.36 ^b^ (±1098.8)	246.13 (±694.62)	***

^1^*p*-Value: ** *p* < 0.01; *** *p* < 0.001; n/s = no significant differences between elevations gradients from 400 to 2000 m.a.s.l; ^a,b^ Letters in superscript denote significant differences among altitudinal gradients.

## Data Availability

This is not applicable as the data are not in any data repository of public access; however, if editorial committee needs access, we will happily provide it. Please use this email: btorres@uea.edu.ec.

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
