# Peer review of "Livelihood Capitals and Opportunity Cost for Grazing Areas’ Restoration: A Sustainable Intensification Strategy in the Ecuadorian Amazon"

_animals, 2023, doi:10.3390/ani13040714_

Round 1

Reviewer 1 Report

The manuscript addresses important and current issues related to livelihood capitals and opportunity cost for grazing areas restoration on the example of the Ecuadorian Amazon.

Please limit the number of self-citations by authors (Bolier Torres and Antón García).

Please consider whether it is necessary to refer to the same literature source more than once in the same paragraph: "Second, land-sharing or wildlife-friendly farming [64], where livestock itself can foster ecosystem services through landscape, triggering ecological [65] and productivity gains in a landscape [64,66,67]".

Please correct Conclusions chapter. Please focus in the Conclusions chapter on what the research in the manuscript brings to science. In the Conclusions chapter, please emphasize the importance of the analyzes carried out. Is it possible to use the research results in practice? What are their limitations of the conducted analyses? Can you indicate further problems to be solved that are related to the topic of the manuscript? Some of the elements I am asking about appeared in the manuscript - please briefly summarize them in the Conclusions chapter and underline the most important elements.

Please adapt the manuscript to the requirements of the journal, e.g. Tables and Figures captions (bold and no dots at the end of the caption).

Author Response

Manuscript ID animals-2160551

Livelihood capitals and opportunity cost for grazing areas restoration: A sustainable intensification strategy in Ecuadorian Amazon

Dear reviewers:

On behalf of all the authors of the article entitled: Livelihood capitals and opportunity cost for grazing areas restoration: A sustainable intensification strategy in Ecuadorian Amazon; I appreciate your kind comments and suggestions, as they have allowed us to improve the scientific quality of the manuscript.  Below we present in detail and by number of lines the changes made to the text:

On the one hand, we are reviewing the manuscript correcting the 9% of similarities found in the manuscript.

Simple summary

Land-sparing and land-sharing should be considered as complementary strategies to favor pasture areas restoration in the Ecuadorian Amazon Region (EAR). Its implementation will depend on factors such as the livelihoods, natural resources valuation and income from livestock activity (dual purpose cattle farms).

A sample of 167 farms of EAR distributed in the three altitudinal gradients (high, medium, and low) was used. The different livelihood capitals and the opportunity cost of the grazing area was calculated.

Therefore, to promote pasture restoration areas policy strategy should aim to maximize net benefits by unit area (ha). Starting from the results obtained, different strategies of restoration are proposed by gradient altitudinal and productive specialization.

  1. a) Land-sparing restoration actions: In middle hill and low zones, land could be destined to produce ecosystem services. A strategy that promotes ecosystem restoration through environmental payments, which considers at least the opportunity and transaction costs.
  2. b) Land-sharing pasture restoration would focus on high mountain farms with a strategy of sustainable intensification and food security.

Reviewer 1

The manuscript addresses important and current issues related to livelihood capitals and opportunity cost for grazing areas restoration on the example of the Ecuadorian Amazon.

Thank you very much, we think it is of great practical interest to relate the livelihood capials with oppoortunity cost. These values can be known from the farm data. This aspect is very new and marks the strategy for an adequate policy of environmental payments. We develop a methodology that can be applied in any context.

Please limit the number of self-citations by authors (Bolier Torres and Antón García).

Excuse us, do not think that it is an exercise in narcissism and Adamism. We want to show the path traveled with previous articles until we can reach this practical application. In the first we have known the system from the perspective of emissions; both in soil, pastures and tree cover. Then we have addressed dairy farms, and as a result of all this we can address this article. Without the previous thing (knowledge of the ecosystem) it is difficult to get to this point.

Please consider whether it is necessary to refer to the same literature source more than once in the same paragraph: "Second, land-sharing or wildlife-friendly farming [64], where livestock itself can foster ecosystem services through landscape, triggering ecological [65] and productivity gains in a landscape [64,66,67]".

has been corrected.

Please correct Conclusions chapter. Please focus in the Conclusions chapter on what the research in the manuscript brings to science. In the Conclusions chapter, please emphasize the importance of the analyzes carried out. Is it possible to use the research results in practice? What are their limitations of the conducted analyses? Can you indicate further problems to be solved that are related to the topic of the manuscript? Some of the elements I am asking about appeared in the manuscript - please briefly summarize them in the Conclusions chapter and underline the most important elements.

Please adapt the manuscript to the requirements of the journal, e.g. Tables and Figures captions (bold and no dots at the end of the caption).

The study shows that opportunity cost, linked to net benefit from grazing area (ha), was a robust, direct, and simple and appropriate indicator. It was used to value the indirect use and hence the ecosystem services of ecosystem restoration. Thus, the opportunity cost could be used as a method to determine public policies for environmental payments and the development of best management practices for sustainable intensification. It will be possible to use the research results in a quantitative approach to environmental payments. Ecosystem services should incorporate at least opportunity cost plus transaction and additional costs. These aspects could be considered in future research and represent a limitation of the study.

Two differentiated strategies are proposed: BMPs in land-sparing are oriented fundamentally in middle hill and lowland zones, promoting ecosystem restoration through environmental payments. Furthermore, BMPs in land-sharing would focus on high mountain farms with a strategy of sustainable intensification and food security.

However, both strategies should complement actions to concentrate conservation efforts especially in areas of high biodiversity (hotspots) conserving as much as possible the ecological dynamics through restoration strategies where it is most appropriate, considering the livelihood outcomes and opportunity cost of grazing areas in pastoral regions.

Reviewer 2 Report

The subject of the paper is current, of high interest and scientific relevance, for which the authors are congratulated for their proposal.

 The following Comments and Suggestions for Authors are added:

Line 24. Correct the acronym Ecuadorian Amazon Region (RAE) by EAR

Introduction. It is recommended to add a paragraph with the description of dual-purpose and intensive livestock production systems according to their respective objectives, sizes, technological levels and levels of marginalization of their producers.

Materials and Methods. Although there is a description of the Pastoral context, it is necessary to include the socioeconomic context of the three study territories, where livelihoods are developed where some indicators of marginality or human development index are included.

Lines 243 and 287, 303. Correct the word “Tabla”

Line 346. It is recommended to add which of the 17 Sustainable Development Goals (SDGs) they refer to

Author Response

Manuscript ID animals-2160551

Livelihood capitals and opportunity cost for grazing areas restoration: A sustainable intensification strategy in Ecuadorian Amazon

Dear reviewers:

On behalf of all the authors of the article entitled: Livelihood capitals and opportunity cost for grazing areas restoration: A sustainable intensification strategy in Ecuadorian Amazon; I appreciate your kind comments and suggestions, as they have allowed us to improve the scientific quality of the manuscript.  Below we present in detail and by number of lines the changes made to the text:

On the one hand, we are reviewing the manuscript correcting the 9% of similarities found in the manuscript.

Simple summary

Land-sparing and land-sharing should be considered as complementary strategies to favor pasture areas restoration in the Ecuadorian Amazon Region (EAR). Its implementation will depend on factors such as the livelihoods, natural resources valuation and income from livestock activity (dual purpose cattle farms).

A sample of 167 farms of EAR distributed in the three altitudinal gradients (high, medium, and low) was used. The different livelihood capitals and the opportunity cost of the grazing area was calculated.

Therefore, to promote pasture restoration areas policy strategy should aim to maximize net benefits by unit area (ha). Starting from the results obtained, different strategies of restoration are proposed by gradient altitudinal and productive specialization.

  1. a) Land-sparing restoration actions: In middle hill and low zones, land could be destined to produce ecosystem services. A strategy that promotes ecosystem restoration through environmental payments, which considers at least the opportunity and transaction costs.
  2. b) Land-sharing pasture restoration would focus on high mountain farms with a strategy of sustainable intensification and food security.

Reviewer 2

The subject of the paper is current, of high interest and scientific relevance, for which the authors are congratulated for their proposal.

Thank you very much, we think it is of great practical interest to relate the livelihood capials with oppoortunity cost. These values can be known from the farm data. This aspect is very new and marks the strategy for an adequate policy of environmental payments. We develop a methodology that can be applied in any context.

 The following Comments and Suggestions for Authors are added:

Line 24. Correct the acronym Ecuadorian Amazon Region (RAE) by EAR

Ok it has been modified.

Introduction. It is recommended to add a paragraph with the description of dual-purpose and intensive livestock production systems according to their respective objectives, sizes, technological levels and levels of marginalization of their producers.

The study was performed in the buffer and transition zone of the SBR. From 464 households distributed in the three zones, those farms with more than 10 heads of cattle and more than three years of consecutive in dual-purpose activity (both milk and meat) were identified. Dual puopose cattle is a mixed system widely distributed in developing countries; characterized by, small size, low productivity, little or no technological level, low level of use of external inputs and with diversified activities (milk, meat, work, crop, etc.) and high level of marginalization of farmers [2,5 ,49,72].Materials and Methods. Although there is a description of the Pastoral context, it is necessary to include the socioeconomic context of the three study territories, where livelihoods are developed where some indicators of marginality or human development index are included.

It has been modified

Lines 243 and 287, 303. Correct the word “Tabla”

It has been modified

Line 346. It is recommended to add which of the 17 Sustainable Development Goals (SDGs) they refer to

It has been modified

low opportunity cost; the land could be destined to the production of ecosystem services (SDG 13, SDG 15) based on the land-sparing approach. The combination of these two strategies favors sustainability [89] and is compatible with the achievement of the 2030 Agenda goals (SDG 13, SDG 15) [54,87,90].

Round 2

Reviewer 1 Report

The manuscript can be accepted in present form.